# Enhancement of Machine-Learning-Based Flash Calculations near Criticality Using a Resampling Approach

**Eirini Maria Kanakaki** [1]⬤, **Anna Samnioti** [1]⬤ **and Vassilis Gaganis** [1,2,*]

[1]  School of Mining and Metallurgical Engineering, National Technical University of Athens, 15780 Athens, Greece; ekanakaki@metal.ntua.gr (E.M.K.); asamnioti@metal.ntua.gr (A.S.)
[2]  Institute of Geoenergy, Foundation for Research and Technology-Hellas, 73100 Chania, Greece
[*]  Correspondence: vgaganis@metal.ntua.gr

**Abstract:** Flash calculations are essential in reservoir engineering applications, most notably in compositional flow simulation and separation processes, to provide phase distribution factors, known as k-values, at a given pressure and temperature. The calculation output is subsequently used to estimate composition-dependent properties of interest, such as the equilibrium phases' molar fraction, composition, density, and compressibility. However, when the flash conditions approach criticality, minor inaccuracies in the computed k-values may lead to significant deviation in the dependent properties, which is eventually inherited to the simulator, leading to large errors in the simulation. Although several machine-learning-based regression approaches have emerged to drastically accelerate flash calculations, the criticality issue persists. To address this problem, a novel resampling technique of the ML models' training data population is proposed, which aims to fine-tune the training dataset distribution and optimally exploit the models' learning capacity across various flash conditions. The results demonstrate significantly improved accuracy in predicting phase behavior results near criticality, offering valuable contributions not only to the subsurface reservoir engineering industry but also to the broader field of thermodynamics. By understanding and optimizing the model's training, this research enables more precise predictions and better-informed decision-making processes in domains involving phase separation phenomena. The proposed technique is applicable to every ML-dominated regression problem, where properties dependent on the machine output are of interest rather than the model output itself.

**Keywords:** phase behavior; machine learning; resampling; flash computations; reservoir simulation; computational thermodynamics

## 1. Introduction

Vapor–liquid equilibrium (VLE) calculations constitute the foundation of a very wide range of reservoir engineering applications [1–3]. These include phase behavior modeling, compositional reservoir simulation, material balance models, Enhanced Oil Recovery (EOR) studies, and separation processes [3]. For instance, in reservoir simulations, given the pressure (P), temperature (T), and mole concentration of each component ($z_i$), phase stability analyses [4] and flash calculations [5] predict, at each discretization block of the grid model (Figure 1) and for each time step, the exact number and type of coexisting phases in equilibrium, as well as the concentration of each component in each equilibrium phase.

The common approach to treating the flash problem, which is depicted in Figure 2, is to seek a solution which satisfies the component mass balance [7–9] and ensures that each component shares same fugacity value in both phases [10]. The principle of mass conservation states that for a fluid with composition **z** that is split into a vapor phase, **y**, and a liquid phase, **x**, the total number of moles of a specific component, $z_i$, is equal to the number of moles of that component in the vapor phase, $y_i$, plus the number of moles of that component in the liquid phase, $x_i$. The equality of fugacities dictates that at equilibrium,

the fugacities of each component in the two phases must be equal, i.e., $f_i^V = f_i^L$, such that the Gibbs free energy of the final two-phase system is minimized [11].

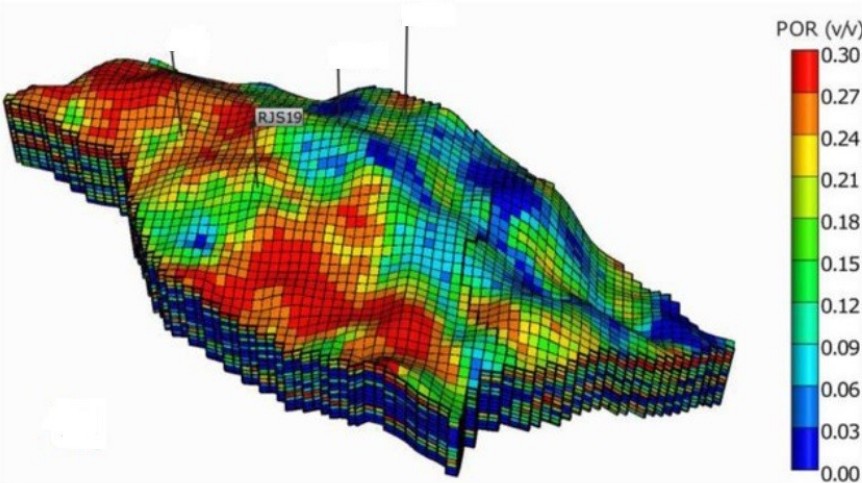

**Figure 1.** Reservoir model consisting of millions of grid blocks [6].

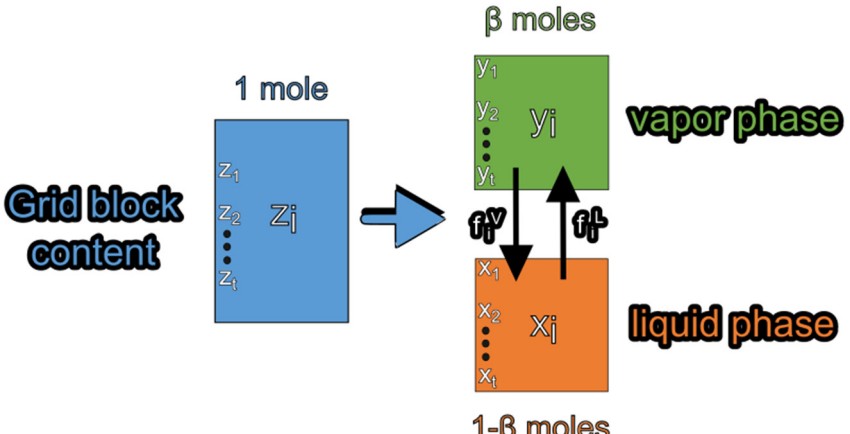

**Figure 2.** Schematic representation of the flash problem.

By combining the two principles and considering the constraint that the composition of each equilibrium phase sums up to unity, the well-known Rachford–Rice equation [12] is derived, which is frequently written as a function of the vapor phase molar fraction, $\beta$:

$$h(\beta) = \sum_i^t \frac{z_i(k_i - 1)}{1 + \beta(k_i - 1)} = 0 \qquad (1)$$

where $z_i$ and $k_i$ denote the overall molar fraction and equilibrium ratio (k-value or distribution factor) of component $i$ in the mixture, respectively. Imposing specific k-values essentially ensures the equality of component fugacities.

The Rachford–Rice equation (Equation (1)) demonstrates the direct effect that equilibrium ratios (k-values) have on the flash calculations. By numerically solving this equation for a given feed, **z**, and values **k**, the molar fraction of the vaporized feed, $\beta$, can be calculated. The composition of each equilibrium phase is then given by:

$$x_i = \frac{z_i}{1 + \beta(k_i - 1)} \qquad (2)$$

$$y_i = \frac{z_i k_i}{1 + \beta(k_i - 1)} = x_i k_i \qquad (3)$$

The range of, which determines the preference of a constituent to remain in the vapor phase or liquid phase, can vary widely depending on the pressure and temperature conditions under which the flash calculation is performed [13]. A component's k-value greater than one indicates that it is volatile and tends to stay mostly in the vapor phase, while a k-value less than one indicates its higher affinity to remain in the liquid phase [14]. As the flash conditions approach the mixture's critical ones (known as criticality conditions), the vapor and liquid phases become less distinguishable from each other, resulting in the k-values converging to unity [15]. Figure 3 provides an illustration of this variation by depicting isotherms plotted on a k-values versus pressure diagram for three components of different molecular weights: a light component ($C_1$), an intermediate component (i-$C_5$), and a heavy component ($C_{7+}$). The k-values clearly approach unity as pressure increases close to the critical one (approx. 4500 psi).

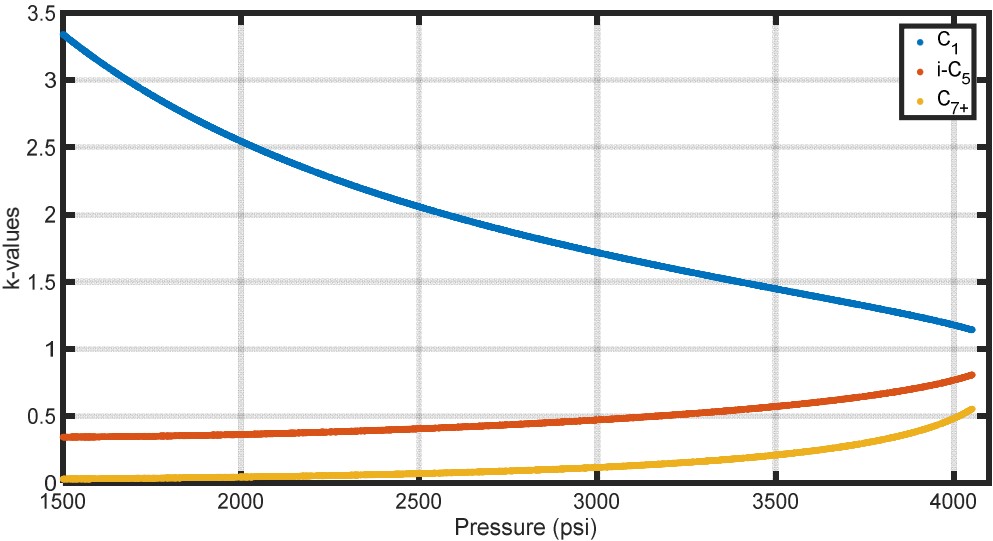

**Figure 3.** Effect of pressure on k-values at reservoir temperature in the 1500 to 4054.3 psi range.

Once the phase compositions are determined, the properties directly incorporated into the flow simulation, such as saturation, density, and viscosity, are computed as functions of $x_i$ and $y_i$.

In the compositional reservoir simulation context, in order to simulate the phase behavior of multicomponent fluids, billions of flash calculations are required to be carried out. Indeed, once the stability test indicates an unstable feed, **z**, at each iteration of the non-linear solver, at each grid block, and at each time step, a flash calculation needs to follow to estimate **k**. Various computational methods are used for flash calculations, including successive substitution iteration (SSI), the quasi-Newton method, the Newton–Raphson (NR), the steepest descent method [16–19], and their respective variations, as well as hybrid approaches [5,20–22]. These computational techniques, though CPU-time intensive, are known for their high accuracy, ensuring precise reservoir simulation predictions across a wide range of pressure–temperature conditions. However, challenges arise when dealing with criticality. This is because properties crucial in flow simulation, such as saturation, density, viscosity, and the effects of gravity, are highly sensitive to k-values under such conditions. Consequently, even minor errors in estimating k-values can result in significant inaccuracies in these important properties.

In addition, several non-iterative methods exist for estimating k-values. Wilson's correlation [23] stands as the most notable, while other correlations include Standing's correlation [24], Hoffman et al.'s correlation [25], Whitson and Torp's correlation [26], and the convergence pressure method [2]. Specific correlations have also been developed for the plus fraction [27–29] and non-hydrocarbons [30]. However, for demanding phase behavior

calculations, the accuracy of these approximations falls short. Therefore, the development of faster and more accurate k-value estimation methods is highly desirable.

Over the years, several machine learning (ML) techniques have appeared in the literature, aiming to accelerate the time-consuming process of solving flash calculations. A phase stability-targeted support vector machine (SVM) methodology was first proposed by Gaganis et al. [31], who utilized a uniformly drawn stability test dataset to generate a discriminating function that replicates a mixture's phase boundary. The classifier was trained using stability labels (stable/unstable) in order to obtain fast predictions. Later, they [3,32] expanded their research and solved both phase stability and phase-split problems by combining SVMs for classification and ANNs for regression in a single prediction system. To further accelerate the calculations, reduced variables were used to shrink the output, back-transforming the ANN predictions to regular k-values. In 2014, Gaganis et al. developed a technique to rapidly solve the multiphase stability problem using SVMs [33]. After that, Gaganis [34] proposed a more efficient treatment of the stability problem utilizing two custom discriminating functions, each single-sided correct, to denote the stability of a mixture. The functions are built so that the ambiguous space, called "the grey area" (where no discriminating function is positive), is as narrow as possible, reducing the need to run a conventional stability test.

Kashinath et al. [35] also treated the stability problem as a binary classification one using SVMs, this time tailoring it to $CO_2$ flooding simulations. In their work, if the classifier predicts an unstable phase, an ANN model is used to predict the prevailing k-values. Zhang et al. [36] introduced a self-adaptive deep learning model capable of predicting the number of phases and their respective properties. Li et al. [37] presented a deep artificial neural network (ANN) model to tackle the iterative flash problem, which is a prevalent issue in phase equilibrium calculations within the moles, volume, and temperature (NVT) framework. Similarly, Poort et al. [38] employed a combination of classification and regression neural networks to address both phase stability and phase property predictions. In addition, Wang et al. [39] built two ANN models to handle the stability and phase-split problems. Similar processes were developed by various other authors [40–43]. Schmitz et al. [44] developed a classification method using a feed-forward ANN and a probabilistic ANN to extend the previous approaches and solve the phase stability problem. A more recent work was developed by Samnioti et al. [45], who employed ANNs to accelerate complex gas condensate phase behavior calculations. The ANN was trained using an extensive dataset obtained from the simulation of various gas recycling schemes, covering any possible compositional changes that might occur inside a reservoir to account for the large compositional variability in the gas reinjection process. Later, Anastasiadou et al. [46] progressed similarly by trying to solve the phase stability problem for an even more complex acid gas reinjection system. The authors proposed three classification approaches, ANNs, decision trees (DTs) and SVMs, to solve the phase stability problem, using a large ensemble of training data.

The main drawback of the ML techniques described above is that the error function utilized in the models' training accounts equally for all datapoints regardless of their proximity to criticality. As a result, when the prevailing conditions are close to the critical ones, as can be the case in gas condensate reservoirs, these models' k-value estimates may lead to significantly large errors in the fluid properties of real interest in flow simulations. It should be noted that apart from the critical point itself, criticality also appears along the convergence locus (CL), that is, the pressure–temperature conditions' locus, where negative flash solutions vanish [47]. In the case of gas condensates, the CL lies very close to the dew point phase boundary and hence to the interior of the phase envelope where the flash calculations are run.

In this paper, we present a novel methodology aimed at enhancing the training quality of ML models addressing the thermodynamic phase-split problem. Our approach focuses on improving the efficiency of ML models, particularly in the vicinity of criticality, by generating uniformly distributed rather than biased deviations across various flash

conditions. Specifically, we propose an approach to fine-tune the ML model's learning capacity without altering its structure or training algorithm while only affecting the training dataset, taking into consideration the impact of k-values on the fluid property of interest in the subsequent flow simulation.

This technique is directly applicable to a wide range of computational problems where an ML model utilizes an input, **x,** to predict an output, $\mathbf{f}(\mathbf{x})$, although the primary focus lies on the accuracy of some dependent property, $\mathbf{g}(\mathbf{f}(\mathbf{x}))$. Our research aligns with the broader field of optimizing regression machines for specific engineering objectives, and it delves into the realm of computational methods designed to improve the performance of these machines in a targeted manner.

The paper is laid out as follows: Section 2 establishes formally the need for a new physics-oriented approach to train ML models that solve the flash problem. Section 3 describes the proposed methodology, while Section 4 discusses the results obtained. Conclusions are presented in Section 5.

## 2. Proof of Concept

In this section, the significance of obtaining poor-quality k-values when running flash calculations close to critical conditions is demonstrated firstly by a theoretical analysis, followed by numerical calculations. In a regular two-phase flash calculation, the quantity of a phase, such as the vapor molar fraction, represented by $\beta$, always falls within the physical interval [0, 1]. This implies that the recombination of $\beta$ moles of gas with composition **y** and $(1 - \beta)$ moles of liquid with composition **x** results in the reconstruction of the original feed composition, Whitson and Michelsen [47] extended the regular flash calculation to conditions under which the fluid is physically monophasic and demonstrated that the phase-split equations can be satisfied even when the $\beta$ values lie outside the physical domain [0, 1]. In a negative flash with $\beta < 0$ (at pressures exceeding the bubble point), $|\beta| = -\beta$ represents the vapor phase amount that needs to be removed from $1 - \beta = 1 + |\beta|$ moles of liquid to reconstruct one mole of the original fluid feed composition. Similarly, when $\beta > 1$ (at pressures above the upper or below the lower dew point), $\beta - 1$ moles of liquid need to be removed from $\beta$ moles of gas. Clearly, the negative flash results are indicative of hypothetical states and lack direct applicability in fluid flow computations. However, such calculations can significantly enhance the convergence properties of regular flash computations near the phase boundary by allowing phase molar fractions at some iteration to temporarily cross the phase boundary.

The convergence pressure ($P_{conv}$) [27] in a multicomponent mixture refers to the pressure at which a negative flash, **k,** approaches unity at a fixed temperature. Similarly, for a fixed pressure, the convergence temperature ($T_{conv}$) is defined as the temperature at which a negative flash, **k,** converges to unity. In the pressure–temperature plane, the CL is the line that connects all of the convergence pressures and temperatures. The regular phase envelope and the CL meet at the mixture's critical point. The negative flash calculations yield non-trivial results, meaning two distinct solutions for the compositions of the liquid and gas phases, only within the region bounded by the regular phase envelope and the convergence locus (CL), which contains the convergence pressures and temperatures. This region is often referred to as the shadow region. The regions discussed are shown in Figure 4 for a gas condensate. Note that the diamond marker represents the fluid's critical point.

Performing flash calculations in the vicinity of a fluid's CL within the phase boundary is challenging as only slightly inaccurate k-value estimates may lead to significant phase compositional errors. This can be expressed mathematically through the limit of $\partial x_i / \partial k_j$ as the k-value of component $j$, $k_j$, approaches unity. Note that $\partial x_i / \partial k_j$ is the partial derivative of Equation (2) with respect to $k_j$ and represents the sensitivity of $x_i$ (or $y_i$) to inaccuracies in $k_j$:

$$\frac{\partial x_i}{\partial k_j} = \frac{-z_i}{[1 + \beta(k_i - 1)]^2} \left[ \beta \delta_{ij} + (k_i - 1) \frac{\partial \beta}{\partial k_j} \right] \tag{4}$$

where $\frac{\partial \beta}{\partial k_j} = \frac{1}{\left(k_j - 1\right)^2}$, and $\delta_{ij}$ is Kronecker's delta. This expression is derived by differentiating the implicit Rachford–Rice equation (Equation (1)). Specifically, the independence of the k-values from each other allows for the cancellation of the summation in Equation (1). By applying the quotient rule and expressing $\partial \beta / \partial k_j$ in terms of the other variables, the partial derivative $\partial \beta / \partial k_j$ is obtained. When $k_j$ approaches unity, as indicated in Equation (5), the limit of $\partial x_i / \partial k_j$ tends towards infinity:

$$\lim_{k_j \to 1} \frac{\partial x_i}{\partial k_j} = \lim_{k_j \to 1} \frac{-z_i}{\left[1 + \beta(k_i - 1)\right]^2} \left( \beta \delta_{ij} + \frac{1}{k_j - 1} \right) = \lim_{k_j \to 1} \left( \beta \delta_{ij} + \frac{1}{k_j - 1} \right) = \infty \quad (5)$$

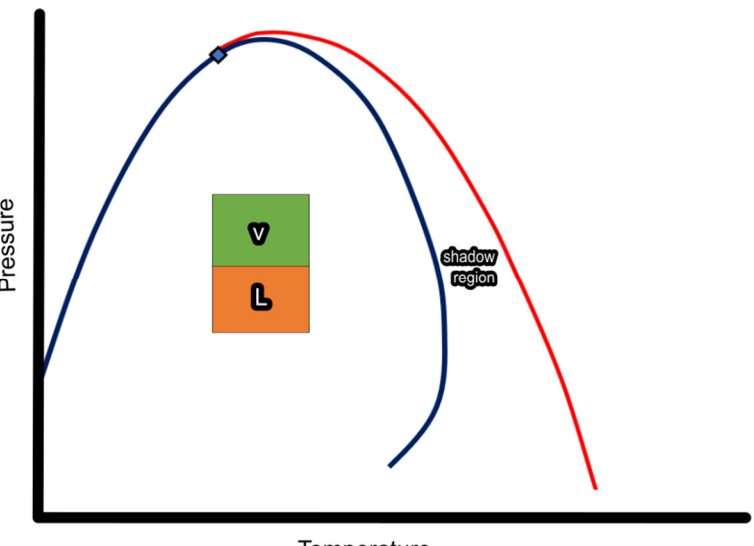

**Figure 4.** Regular phase envelope and shadow region of a gas condensate.

Thus, when the prevailing conditions are close to criticality (either to the critical point itself or to the CL), the need for precise k-value, **k**, estimates is particularly high to ensure that dependent properties of real interest in reservoir simulation, such as the saturation, density, and viscosity of each phase, are computed accurately. Densities are directly related to gravitational effects, whereas their derivatives with respect to pressure are directly related to fluid compressibility, which dominates viscous flow.

After mathematically demonstrating the extent of the k-values' limited accuracy problem, numerical calculations were also carried out on two real reservoir fluids, a lean gas condensate and a rich gas condensate, at various sets of pressure–temperature (P–T) conditions. The compositions of the utilized fluids are shown in Table 1. Figure 5 illustrates the phase envelopes (depicted by the blue curves) of the lean gas condensate and the rich gas condensate, respectively, as obtained using the Peng and Robinson cubic equation of state (1978) [2,14,48–53]. The diamond-shaped blue points on the phase envelopes represent the critical points of the mixtures, while the purple dashed lines represent the convergence loci (CLs). Five points were selected along the red isotherm of the reservoir temperature of each fluid, each exhibiting a varying distance to criticality.

**Table 1.** Composition of gas condensates in mole% and reservoir temperature.

| Comps | Lean Gas Condensate | Rich Gas Condensate |
|:---:|:---:|:---:|
| $N_2$ | 0.01 | 0.10 |
| $CO_2$ | 1.26 | 2.95 |
| $C_1$ | 80.22 | 72.18 |
| $C_2$ | 6.6 | 4.03 |
| $C_3$ | 3.25 | 4.78 |

**Table 1.** *Cont.*

| Comps | Lean Gas Condensate | Rich Gas Condensate |
|---|---|---|
| i-C$_4$ | 0.73 | 1.44 |
| n-C$_4$ | 1.08 | 2.38 |
| i-C$_5$ | 0.47 | 1.10 |
| n-C$_5$ | 0.38 | 0.99 |
| C$_6$ | 0.63 | 1.28 |
| C$_{7+}$ | 5.37 | 8.77 |
| **T$_{res}$ (°F)** | 262 | 268 |

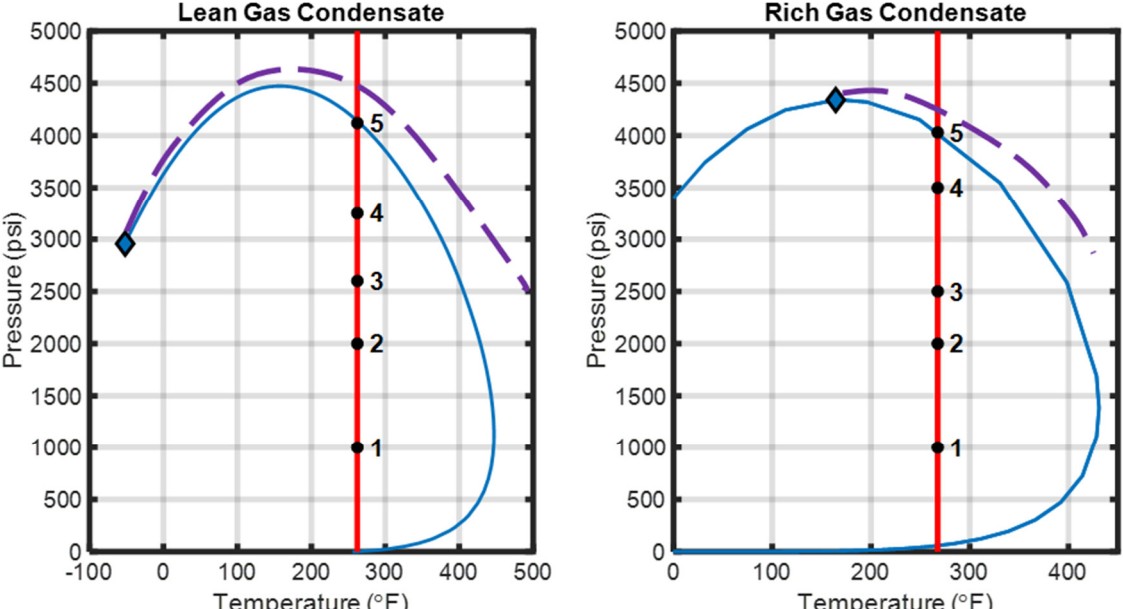

**Figure 5.** Phase envelope and convergence locus of the lean (on the **left**) and rich (on the **right**) gas condensates.

The k-value norm, $\mathcal{L}_k$, accounting for the sum of the squares of the natural logarithms of the experimental k-values, **k** (Equation (6)), was introduced to serve as an indicator of a point's proximity to criticality, with lower values denoting a closer proximity. From Tables 2 and 3, it follows that among the five points selected for each fluid (Figure 4), the proximity to criticality increases gradually between the first and fifth point.

$$\mathcal{L}_k = \sum_{i=1}^{t} (log(k_i))^2 \tag{6}$$

**Table 2.** Proximity of the five selected P–T points to criticality—lean gas condensate.

| P–T Points | 1 | 2 | 3 | 4 | 5 |
|---|---|---|---|---|---|
| $\mathcal{L}_k$ | 108.41 | 71.06 | 49.79 | 30.02 | 8.63 |

**Table 3.** Proximity of the five selected P–T points to criticality—rich gas condensate.

| P–T Points | 1 | 2 | 3 | 4 | 5 |
|---|---|---|---|---|---|
| $\mathcal{L}_k$ | 28.59 | 17.88 | 12.99 | 4.55 | 0.80 |

Regular flash calculations were conducted using conventional iterative algorithms to compute the gas phase ratio, $\beta$, and the equilibrium phases' properties at each point along

the isotherm. Subsequently, random noise of a fixed amplitude was added to the k-values to replicate the error of an ML model trained to predict those outputs. This choice stems from the fact that, in this work, an attempt is made to minimize the way an ML model operates. In this case, the absolute error objective function is minimized by the training process instead of a relative error metric. In fact, noise was added to the property predicted by the ML model, which is the logarithm of the k-value. The Rachford–Rice equation was then rerun using the distorted k-values, and the molar ratio and phase properties were reestimated. Finally, the obtained deviations, defined by their average errors, were determined. Tables 4 and 5 demonstrate that as the CL is being approached, there is a significant increase in the absolute errors of the vapor phase molar fraction, $\beta$, the liquid and vapor phase compositions, $\mathbf{x}$ and $\mathbf{y}$, and the liquid and gas phase densities, $\rho_L$ and $\rho_V$, even though the same level of noise was added to all the selected points. This underscores that highly reliable k-value estimations are especially crucial in close proximity to the CL due to the increased sensitivity of the derived phase properties to k-value errors in that area. Note that, for this specific application, adding a relative basis noise would cause an unbalanced effect on the k-values due to their very wide span, which covers several orders of magnitude. This, in turn, would not allow for the analysis of the side effect on the properties of interest for the subsequent flow simulations.

**Table 4.** Absolute errors of direct and dependent properties—lean gas condensate.

| P–T Points | 1 | 2 | 3 | 4 | 5 |
|:---:|:---:|:---:|:---:|:---:|:---:|
| **Proximity to CL** | | Low | Medium | | High |
| | | | Absolute Error | | |
| $\beta \times 10^{-4}$ | 1.18 | 4.59 | 9.95 | 20.8 | 83.1 |
| $\mathbf{x} \times 10^{-3}$ (mole%) | 2.87 | 4.49 | 5.79 | 6.79 | 8.80 |
| $\mathbf{y} \times 10^{-4}$ (mole%) | 1.27 | 2.31 | 4.01 | 7.55 | 24.2 |
| $\rho_L \times 10^{-1}$ (lb/ft$^3$) | 0.35 | 0.97 | 16.5 | 24.7 | 43.5 |
| $\rho_V \times 10^{-2}$ (lb/ft$^3$) | 0.05 | 5.51 | 11.6 | 19.4 | 23.6 |

**Table 5.** Absolute errors of direct and dependent properties—rich gas condensate.

| P–T Points | 1 | 2 | 3 | 4 | 5 |
|:---:|:---:|:---:|:---:|:---:|:---:|
| **Proximity to CL** | | Low | Medium | | High |
| | | | Absolute Error | | |
| $\beta \times 10^{-4}$ | 2.11 | 7.83 | 13.3 | 53.9 | 301 |
| $\mathbf{x} \times 10^{-3}$ (mole%) | 2.41 | 3.92 | 4.60 | 6.94 | 10.9 |
| $\mathbf{y} \times 10^{-4}$ (mole%) | 2.85 | 4.98 | 6.91 | 18.1 | 57.6 |
| $\rho_L \times 10^{-1}$ (lb/ft$^3$) | 0.20 | 0.57 | 0.82 | 1.77 | 3.35 |
| $\rho_V \times 10^{-2}$ (lb/ft$^3$) | 0.15 | 6.24 | 12.1 | 28.5 | 34.1 |

　　To further demonstrate the problem, the efficacy of a classic supervised ML model trained to accurately reproduce the training data associated with the P–T points near the CL of the rich gas condensate was investigated. Firstly, a total of 100,000 random pressure points were uniformly selected across the range of 1500 psi (i.e., a typical abandonment pressure) to 4054.3 psi (i.e., the gas dew point pressure) at the reservoir temperature, and a dedicated MATLAB code that performs regular flash calculations was run for each of these pressures at the reservoir temperature using the feed fluid composition, $\mathbf{z}$, to yield the vapor phase and liquid phase compositions, $\mathbf{y}$ and $\mathbf{x}$, respectively. Subsequently, the corresponding k-values, $\mathbf{k}$, for each pressure point were determined by calculating the ratio of the vapor phase composition of a given component, $i$, to its corresponding liquid phase composition. The data collected form the "source dataset".

　　A "base case dataset" was generated by randomly selecting 2000 uniformly drawn pressure points, along with their associated k-values, $\mathbf{k}$, from the source dataset in order to

ensure statistical significance with respect to the population. The pressure histogram of the base case dataset in Figure 6 exhibits uniformly distributed bars with no discernible peaks or valleys, which confirms its uniformity. On the other hand, the frequency distribution of the corresponding k-value norms is highly non-uniform, as shown in the same figure, as the effect of pressure on the k-values of each component of a mixture at a constant temperature is distinct and depends on the unique properties of each component.

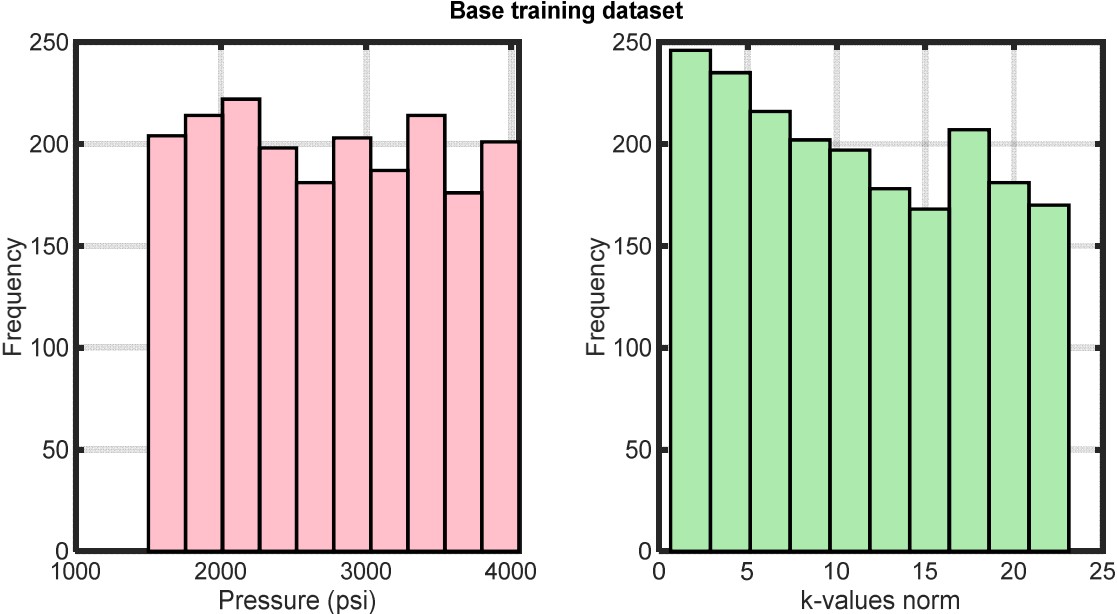

**Figure 6.** Pressure and k-value histograms of the base training dataset.

Subsequently, a conventional feedforward artificial neural network (ANN) was trained against the base case training dataset to predict the logarithms of the k-values, given the prevailing pressure. Note that these pressure values reflect the dynamic conditions within the reservoir, and the role of the ANN is to establish a functional relationship between these pressure inputs and the resulting k-values. Training was repeated 100 times to mitigate the inherent stochasticity of ANN training, which stems from random weight initialization and the stochastic nature of the training algorithm. The Rachford–Rice equation (Equation (1)) was solved for $\beta$ using a combined Newton–Raphson bisection method, given the overall mole fractions, $\mathbf{z}$, and predicted k-values. Given $\beta$, the phase compositions $\mathbf{x}$ and $\mathbf{y}$ were obtained using Equations (2) and (3), respectively, and the molecular weights of the two phases in equilibrium were determined based on their respective compositions. Finally, the Peng–Robinson cubic equation of state was used to compute the molar volumes, $v_m$, of each phase, which, when combined with the molecular weights, enable the liquid and gas densities to be determined. To gain a more comprehensive understanding of the effectiveness of the classic supervised ML model training approach, the properties of interest were averaged across the 100 training runs, yielding a single representative value for each property.

Conventional ANN training aims at utilizing its flexibility to vary the model parameters (weights) to optimally reproduce the training outputs, i.e., the k-values, by minimizing the loss function, $\mathcal{J}$, described by Equation (7).

$$\mathcal{J} = \frac{1}{N} \sum_{i} \sum_{j} \left( \hat{k}_{ij} - k_{ij} \right)^2 \tag{7}$$

where $\hat{k}_{ij}$ and $k_{ij}$ correspond to the estimated and exact k-value, respectively, of the $i$th component of the $j$th pair. As a result, it focuses on accurately reproducing k-values rather than the dependent properties of interest in a flow simulation.

To evaluate the accuracy of the predicted dependent properties in conjunction with their proximity to criticality, the 2000-datapoint training space was divided into 10 classes based on the k-value norms, where the first class encompasses the P–T points that lie closest to the CL, and the last class comprises the points that are farthest from tit. As can be seen from the k-value norm distribution in Figure 5 and the ANN prediction error statistics in Table 6, class 1 contains 246 points, whereas class 10 has only 169 points. From Table 6, it can be further seen that the errors associated with points in close proximity to criticality are significantly greater than those for points that lie further away, although the former dominate the training process due to their abundance in the dataset, resulting in the error function being minimized by the learning step. This is visually confirmed by the decaying value of the average absolute error of all properties with increasing class number, i.e., while departing from criticality (Figure 7). These findings confirm the need for a new, focused approach to training an ANN to solve the thermodynamic phase-split (flash) problem. Moreover, these findings can be repeated using any alternative machine learning model, rather than an ANN, to predict k-values given the flash conditions.

**Table 6.** Absolute average errors in conventional ML model training.

| Class | Range of $\mathcal{L}_k$ | | Num of Points | $\beta \times 10^{-5}$ | $x \times 10^{-4}$ | $y \times 10^{-5}$ | $\rho_L \times 10^{-3}$ (lb/ft$^3$) | $\rho_V \times 10^{-3}$ (lb/ft$^3$) |
|---|---|---|---|---|---|---|---|---|
| **1** | 0.66 | 2.89 | 246 | 180 | 7.66 | 44.2 | 23.4 | 15.8 |
| **2** | 2.92 | 5.14 | 235 | 34.5 | 3.42 | 15.8 | 9.14 | 5.38 |
| **3** | 5.15 | 7.36 | 216 | 18.7 | 3.03 | 9.47 | 7.47 | 2.99 |
| **4** | 7.39 | 9.62 | 202 | 12.3 | 2.48 | 7.67 | 5.55 | 2.20 |
| **5** | 9.65 | 11.87 | 197 | 9.15 | 2.34 | 5.52 | 4.89 | 1.46 |
| **6** | 11.87 | 14.08 | 178 | 7.36 | 2.29 | 5.56 | 4.38 | 1.29 |
| **7** | 14.12 | 16.35 | 168 | 5.25 | 1.80 | 4.34 | 3.16 | 0.88 |
| **8** | 16.37 | 18.58 | 207 | 4.54 | 1.81 | 3.60 | 2.99 | 0.67 |
| **9** | 18.62 | 20.84 | 181 | 4.98 | 2.25 | 4.93 | 3.38 | 0.79 |
| **10** | 20.85 | 23.07 | 169 | 4.37 | 2.20 | 4.71 | 2.99 | 0.63 |

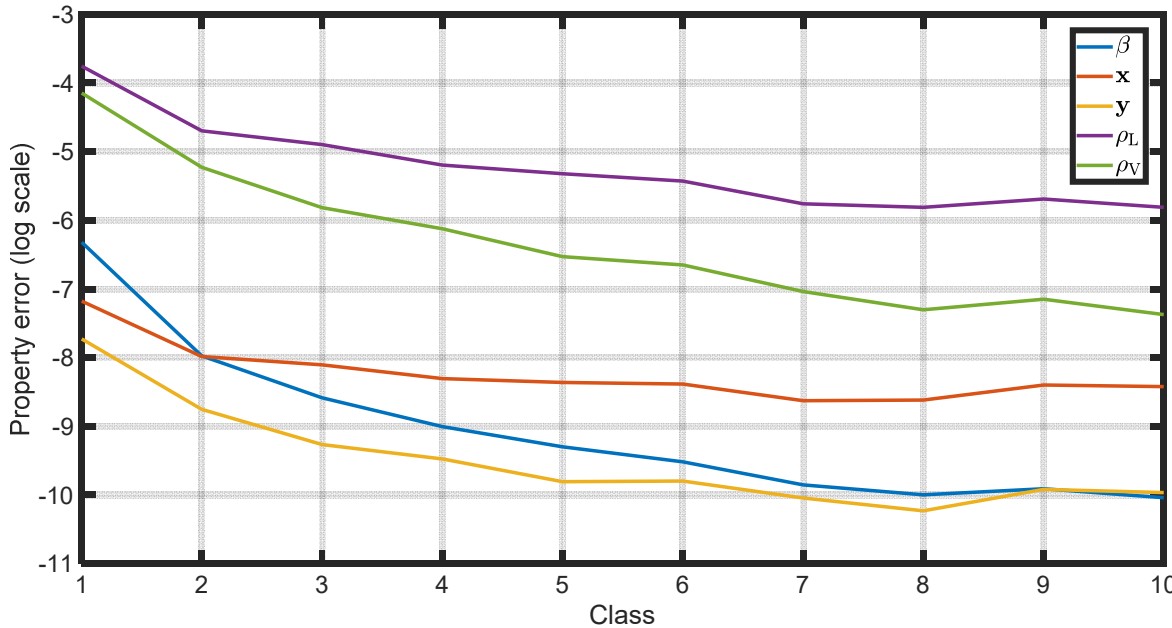

**Figure 7.** Average absolute errors in conventional ML model training for each dependent property and datapoint class.

## 3. Methodology

The previous section established the necessity of focusing on the physical properties of interest, rather than solely on the model output, while training machine learning models. Therefore, ML model training must be modified to prioritize the primary objective, that is, to accurately predict a dependent property, $\mathbf{g}(\mathbf{f}(\mathbf{x}))$, based on an input, $\mathbf{x}$, where $\mathbf{f}(\mathbf{x})$ represents the original ML model output.

To attain a physically sound predictive model, the loss function, $\mathcal{J}$, in Equation (7) can be modified to incorporate weighting factors for individual training points or groups of points. These weights assign varying levels of importance to the datapoints based on their significance in predicting the dependent properties of interest. In the context of this research, highly important datapoints are those in the vicinity of the mixture's convergence locus (CL), where the uncertainty in the indirectly derived properties is maximized. However, implementing this approach requires significant modifications to the training algorithm, such as adjusting the loss function, its gradient, and the Hessian matrix, all of which are indispensable parts of the training algorithm. Instead of introducing individual weights for each datapoint or groups of points, which can be both complex and significantly alter the native training formulation, this work proposes implementing a resampling technique to enhance the flash ML models' predictions near criticality while simultaneously reducing the average error and standard deviation for each property ($\beta$, $\mathbf{x}$, $\mathbf{y}$, $\rho_L$, and $\rho_V$).

The proposed resampling technique is designed to improve the performance of an ML model by balancing the datapoint population within a training dataset derived from a source dataset that encompasses a substantial volume of datapoints. This is accomplished by considering the datapoints' proximity to criticality ($\mathcal{L}_k$) and the prediction errors associated with a specific property of interest within each class of the training dataset. The population of datapoints belonging to classes which exhibit poor performance is enhanced by picking more training samples from the source dataset, corresponding to a stronger contribution of such points to the training error function. This way, the training algorithm that minimizes $\mathcal{J}$ is forced to pay more attention to those points, thus improving their prediction over the other classes and recovering the required accuracy of the physical properties which follow. A hyperparameter, denoted by $D$, controls the level of adjustment made to the number of datapoints in each class during the resampling process. A high value of $D$ results in more intense adjustments, while a lower value results in more subtle changes. $D$ can be considered analogous to the weight given to each class's average logarithmic error of the selected dependent property when balancing the training data. Once the resampling step from the source dataset has been completed, a balanced dataset emerges, and the machine training is run regularly.

Figure 8 outlines the resampling algorithm used to improve the ML model performance against some specific dependent property. Firstly, regular training is performed using the initially available training population to obtain the training error of each datapoint, $\Delta_i$. The algorithm utilizes the average absolute error per class, $e_i$ (Equation (8)), for the derived property of interest, obtained by considering the total number of datapoints in each class, $C_i$, of the training dataset, denoted by $N_i$. Subsequently, the algorithm calculates the log absolute error per class, $\varepsilon_i$ (Equation (9)), with which the extent ($\delta_i$) to which these errors deviate from the corresponding errors of the best-trained class can be determined, as described in Equation (10). Ideally, all classes would share the same average absolute error, leading to $\delta_i$ values equal to zero and hence no need to modify the overall balance of the dataset. For non-zero $\delta_i$ values, the algorithm defined increases the number of samples in each class, as outlined in Equation (11). Clearly, the bigger the spread between a class error, $\varepsilon_i$, and that of the optimally learned one ($\min(\varepsilon_j)$), the bigger the increase in the datapoints in that class, from $N_i$ to $F_i$. In the final step, the resampling algorithm employs uniform down sampling to reduce the number of datapoints in the resampled training dataset while maintaining the new distribution of the resampled training dataset. This process ensures

that the resampled training dataset is of equal size to the base training dataset, preventing any bias introduced by the resampling process.

$$e_i = \frac{\sum_{j \in c_i} |\Delta_j|}{N_i} \tag{8}$$

$$\varepsilon_i = log(e_i) \tag{9}$$

$$\delta_i = D[\varepsilon_i - min(\varepsilon_i)] \tag{10}$$

$$F_i = exp(\delta_i)N_i \tag{11}$$

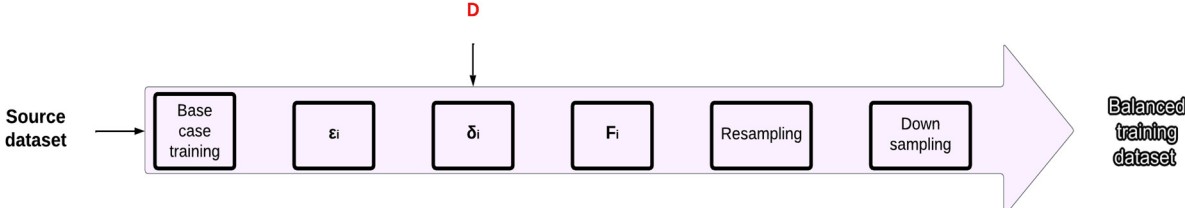

**Figure 8.** Outline of the proposed resampling algorithm.

## 4. Results and Discussion

The proposed resampling technique was applied to the base case training dataset of the rich gas condensate presented in Section 2. Firstly, the liquid density error was selected as the balancing metric to resample the datapoint population. This selection was motivated by the significant liquid dropout that occurs when the pressure of the rich gas condensate falls below the dew point, which needs to be modelled accurately due to its high commercial value. In addition, liquid density is included in the expression of isothermal compressibility, as described in Equation (12), which, in turn, governs fluid flow.

$$c_o = \frac{1}{\rho_L} \left( \frac{\partial \rho_L}{\partial P} \right)_T \tag{12}$$

To determine the optimal value for hyperparameter $D$, which controls the level of adjustment imposed, a sensitivity analysis was performed. This involved varying the value of $D$ across a wide range and evaluating its impact on predicting the five dependent properties of interest ($\beta$, $\mathbf{x}$, $\mathbf{y}$, $\rho_L$, and $\rho_V$) using the ML-model-predicted k-values. As shown in Figure 9, higher values of $D$ significantly adjust the datapoints distribution in each class during resampling, focusing more on classes with poorer performances (i.e., class 1), as reflected by the stronger contribution of such points to the training error function. As an example, consider the distribution of the balanced dataset's population across the 10 classes for $D$ values ranging between 0.8 and 2.4, as obtained by considering the liquid density error (i.e., $e_i$) to control the resampling process. Since the points in each class exhibit varying degrees of proximity to criticality, their cardinality is differently affected by the resampling process. Class 1 contains the datapoints which lie closer to criticality conditions; they exhibit maximum error, and hence their number is severely affected by the correction factor $D$. Specifically, this class originally contained 246 points, which increased to 435 and 697 for $D = 1$ and $D = 2$, respectively, whereas the number of points in the remaining classes decreased accordingly.

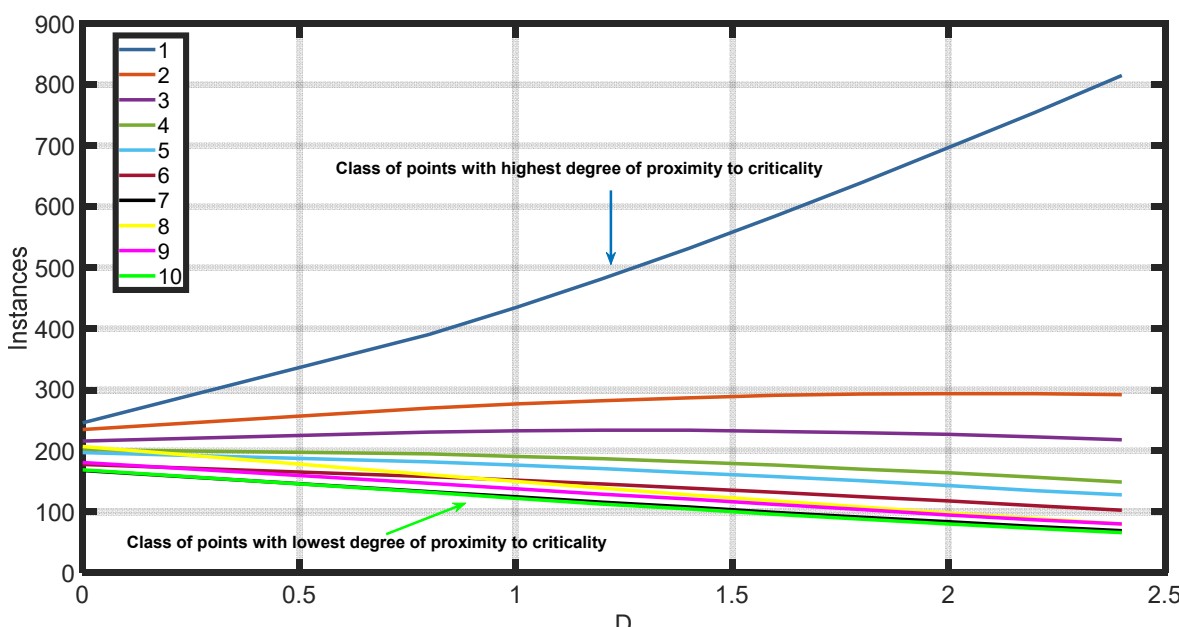

**Figure 9.** Number of datapoints in the balanced dataset across the 10 classes with respect to hyperparameter $D$, based on $\rho_L$ error resampling.

Figure 10 depicts the overall improvement (or decline) in the average error and standard deviation for the combined dependent properties of interest. The k-values utilized in this analysis were generated from the ML model trained with the balanced training dataset of the $\rho_L$ error resampling algorithm. Subsequently, the standard deviation and absolute error of each property for various $D$ values were recorded. The sum of the standard deviations and absolute errors of all five dependent properties at each $D$ was then calculated to obtain a measure of the improvement in predictions over all the properties of interest. Comparing those values to the performance of the original, unbiased model led to the datapoints plotted in Figure 10. The analysis covers a range of $D$ values spanning from 0.8 to 2.4, presenting a comparative evaluation against the results obtained from the base ML training. Clearly, the overall improvement (or decline) in the average error and standard deviation is equal to zero when $D = 0$, i.e., when no resampling takes place.

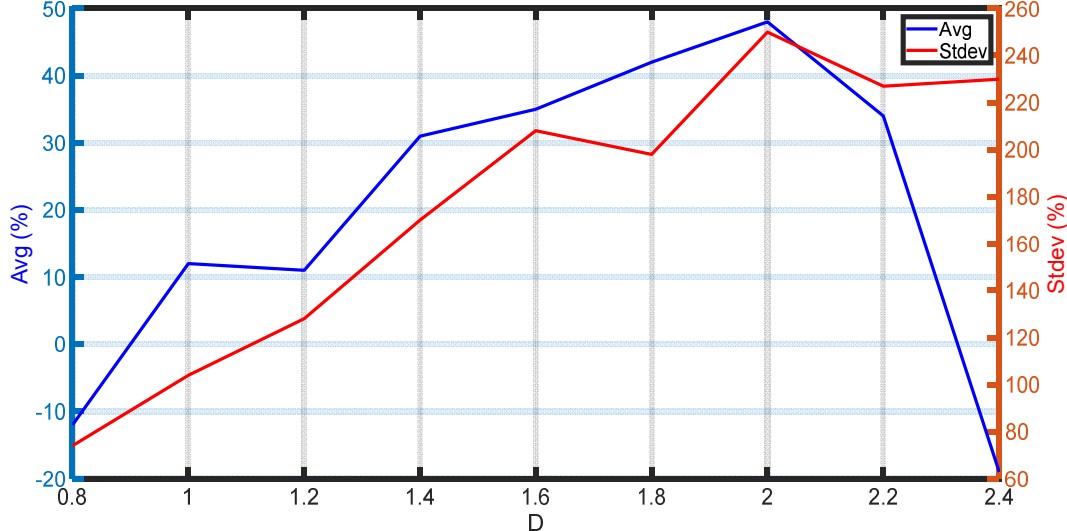

**Figure 10.** Improvement/decline in the average absolute error and standard deviation versus $D$ based on $\rho_L$ error resampling.

Exaggerated values of *D* may lead to overfitting in favor of the formerly weak class, ultimately deteriorating the overall model performance, as reflected by the negative mean absolute error values. This corresponds to an improvement in the errors for classes near criticality, while the deterioration in other classes suggests a negative overall trade-off. For the rich gas condensate base training dataset considered in this work, the optimal value of *D* was found to be equal to two. At this value, the average error and standard deviation per property exhibit substantial improvements, reaching a cumulative improvement of 48% and 250%, respectively, compared to the base case training scenario.

Figure 11 provides a visual representation of the distribution frequency of k-value norms within the optimally resampled training dataset (*D* = 2). As expected, the histogram of k-values exhibits a right-skewed pattern, confirming that a significant portion of the datapoints in the dataset correspond to pressures close to the CL of the fluid. Likewise, the histogram of pressures will display a left-skewed pattern.

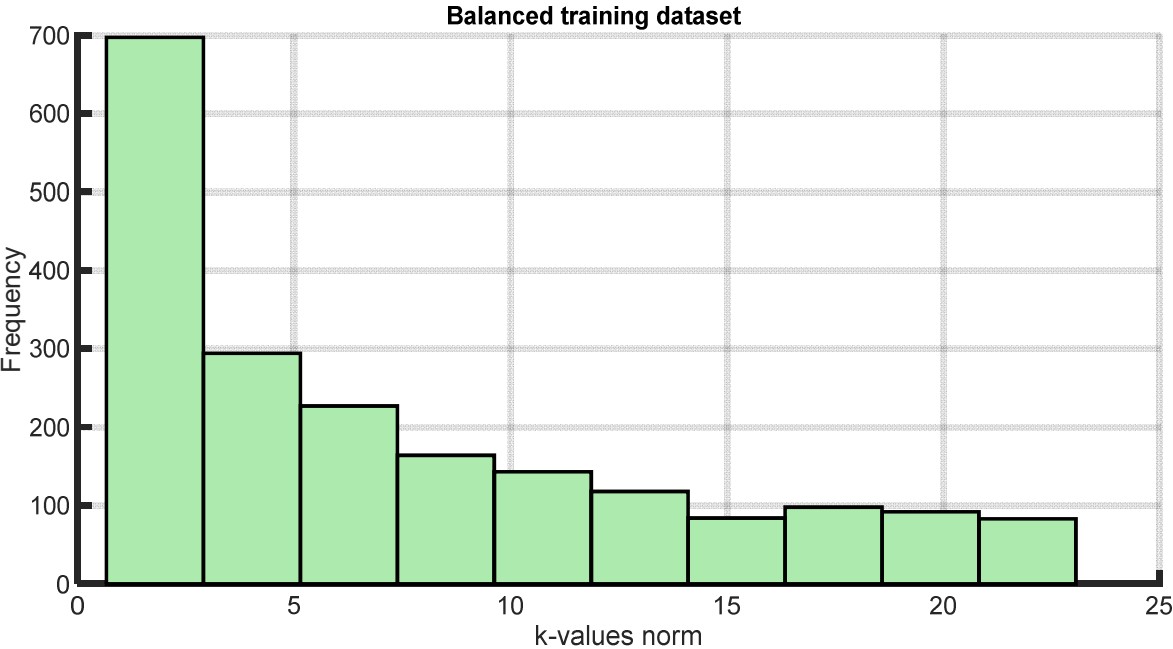

**Figure 11.** K-value histograms of the balanced training dataset.

Table 7 illustrates the absolute average errors within each class obtained from training the ANN using the resampling approach, while Figures 12–16 provide a comparison of the class errors obtained by conventional and balanced dataset training. Based on these results, it is safe to say that the proposed methodology leverages more efficiently the learning capacity of the ANN, leading to significant improvements in the errors of the underperforming classes while reducing, at the same time, the average error and standard deviation for each property. Eventually, the model prediction error in the dependent properties of interest is much more uniformly distributed, thus ensuring the similar performance of the ML model across all flash conditions, only weakly related to their proximity to criticality. Note that the model trained with the resampled training dataset exhibits slightly worse performance for classes 5–10 compared to the model trained with the base training dataset. However, this discrepancy is a strategic trade-off in the approach. Emphasis is put on optimizing the model's learning capacity across various flash conditions, particularly in classes closer to criticality, where inaccuracies in predicted k-values have a more pronounced impact on the results, i.e., the dependent properties. In other words, it is preferable to "sacrifice" some of the performance of the fine-performing classes while improving that of the classes performing really poorly, as is the case with the near-critical points.

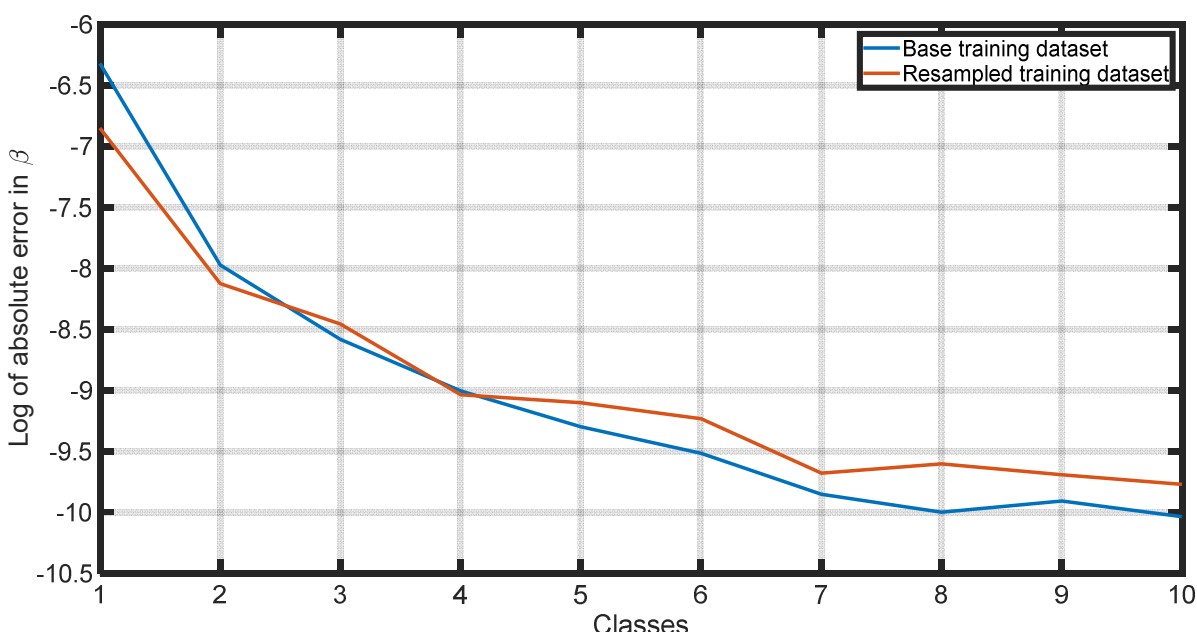

**Figure 12.** Comparison of absolute average error in $\beta$ per class using the base training dataset and the resampled training dataset.

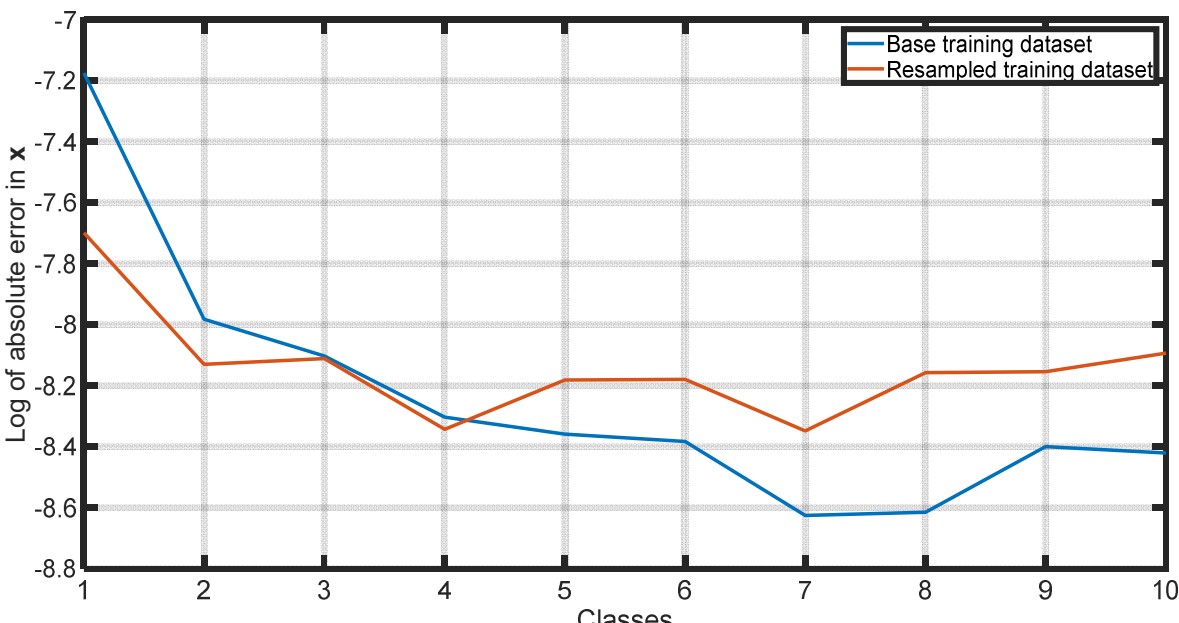

**Figure 13.** Comparison of absolute average error in **x** per class using the base training dataset and the resampled training dataset.

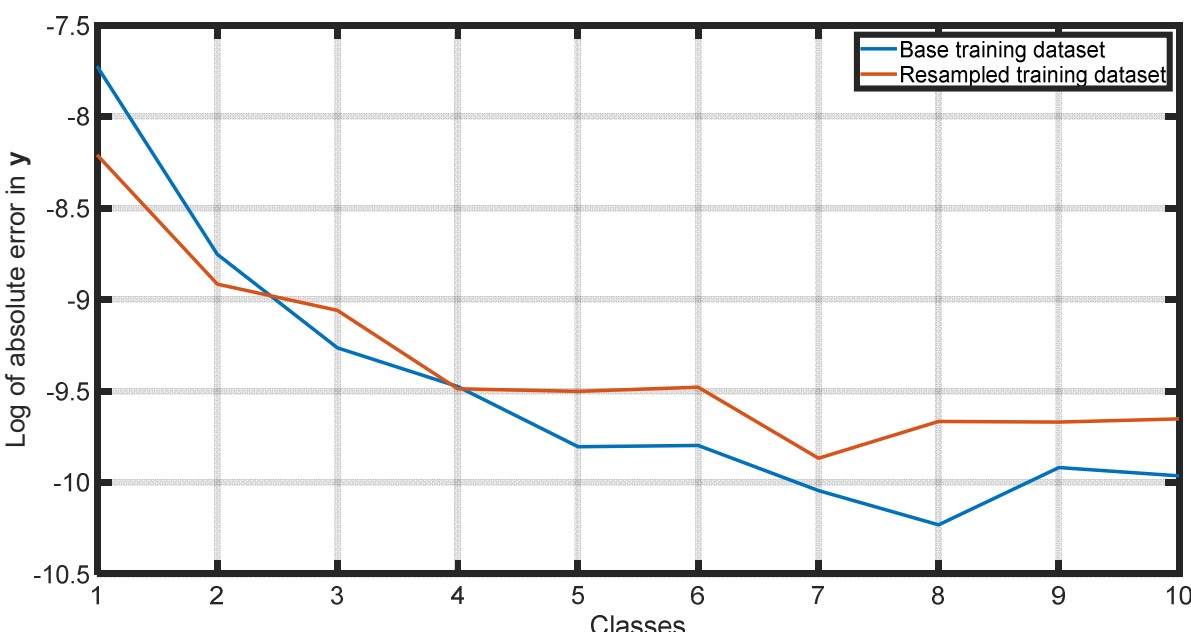

**Figure 14.** Comparison of absolute average error in **y** per class using the base training dataset and the resampled training dataset.

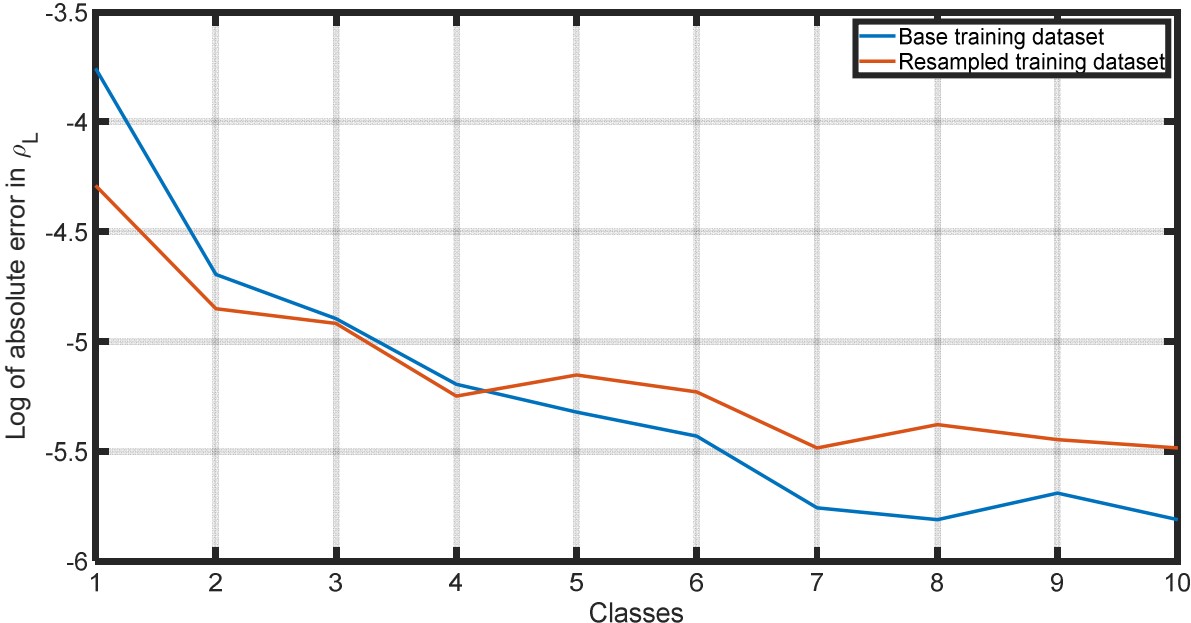

**Figure 15.** Comparison of absolute average error in $\rho_L$ per class using the base training dataset and the resampled training dataset.

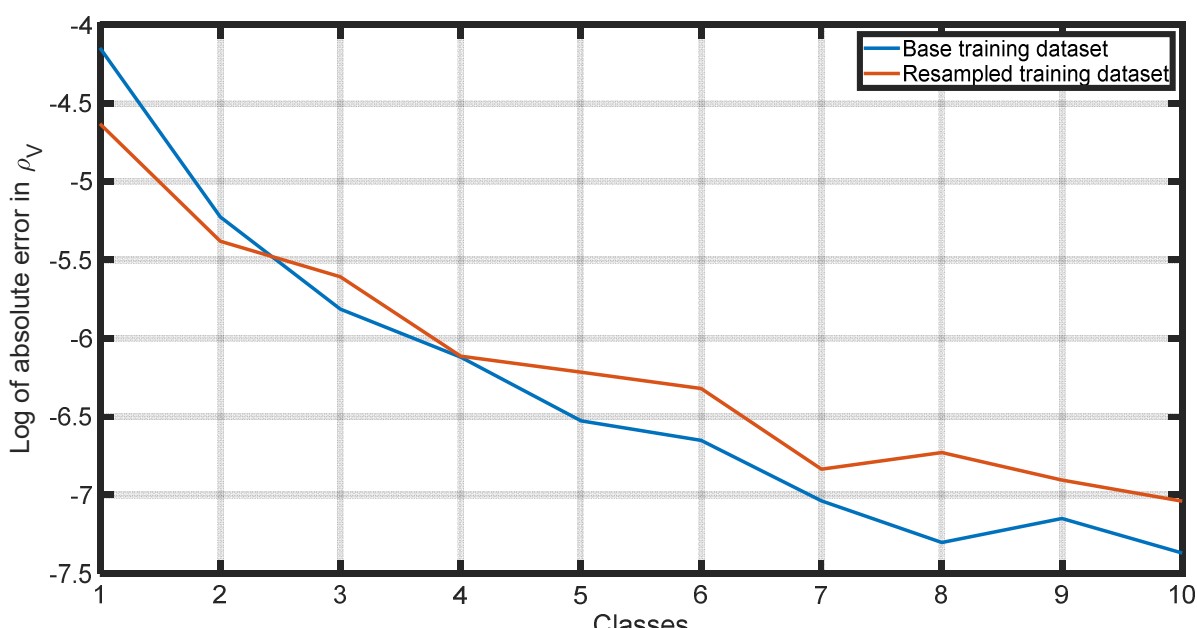

**Figure 16.** Comparison of absolute average error in $\rho_V$ per class using the base training dataset and the resampled training dataset.

**Table 7.** Absolute average errors in resampling ML model training.

| Class | Range of $\mathcal{L}_k$ | | Num of Points | $\beta \times 10^{-5}$ | $x \times 10^{-4}$ | $y \times 10^{-5}$ | $\rho_L \times 10^{-3}$ (lb/ft$^3$) | $\rho_V \times 10^{-3}$ (lb/ft$^3$) |
|---|---|---|---|---|---|---|---|---|
| 1 | 0.66 | 2.89 | 697 | 106 | 4.53 | 27.2 | 13.7 | 9.73 |
| 2 | 2.92 | 5.14 | 294 | 29.6 | 2.95 | 13.4 | 7.82 | 4.60 |
| 3 | 5.15 | 7.36 | 227 | 21.3 | 3.00 | 11.6 | 7.31 | 3.67 |
| 4 | 7.39 | 9.62 | 164 | 11.9 | 2.38 | 7.57 | 5.25 | 2.21 |
| 5 | 9.65 | 11.87 | 143 | 11.1 | 2.80 | 7.47 | 5.79 | 2 |
| 6 | 11.87 | 14.08 | 118 | 9.78 | 2.80 | 7.64 | 5.36 | 1.80 |
| 7 | 14.12 | 16.35 | 84 | 6.26 | 2.37 | 5.19 | 4.15 | 1.07 |
| 8 | 16.37 | 18.58 | 98 | 6.74 | 2.87 | 6.34 | 4.61 | 1.19 |
| 9 | 18.62 | 20.84 | 95 | 6.17 | 2.87 | 6.32 | 4.31 | 1 |
| 10 | 20.85 | 23.07 | 80 | 5.71 | 3.06 | 6.42 | 4.15 | 0.88 |

By reducing the average error, the ANN improves its ability to make predictions that are, on average, closer to the exact values. Additionally, by reducing the standard deviation, the model ensures that the datapoints are less spread out, resulting in more concentrated and reliable predictions. It is interesting to note that although the resampling process was carried out against the liquid phase density solely, the performance of all the dependent properties was positively affected due to their natural correlation.

A second attempt was carried out to resample the dataset, this time based on the error of the vapor phase molar fraction, $\beta$, per class, which determines the saturation of the coexisting phases in equilibrium. Figure 17 depicts the corresponding improvement or decline in the sum of the absolute average errors and standard deviations per property when varying hyperparameter $D$ in the range of 0.8 to 2. The maximum improvement in the overall average error and standard deviation combined occurs when $D$ equals 1.2, which is a 19% reduction in the overall average error and a 245% improvement in the standard deviation. It is important to note that when the error in $\rho_L$ was the basis for the resampling algorithm, the overall improvement in the average error and standard deviation was more

pronounced. This indicates that for each fluid, there is a specific dependent property whose error should be incorporated into the resampling algorithm to optimize predictions using an ML model. Nevertheless, the whole process can be easily automated in a computer program, thus minimizing the use of human resources and the time required to optimize the ML training step.

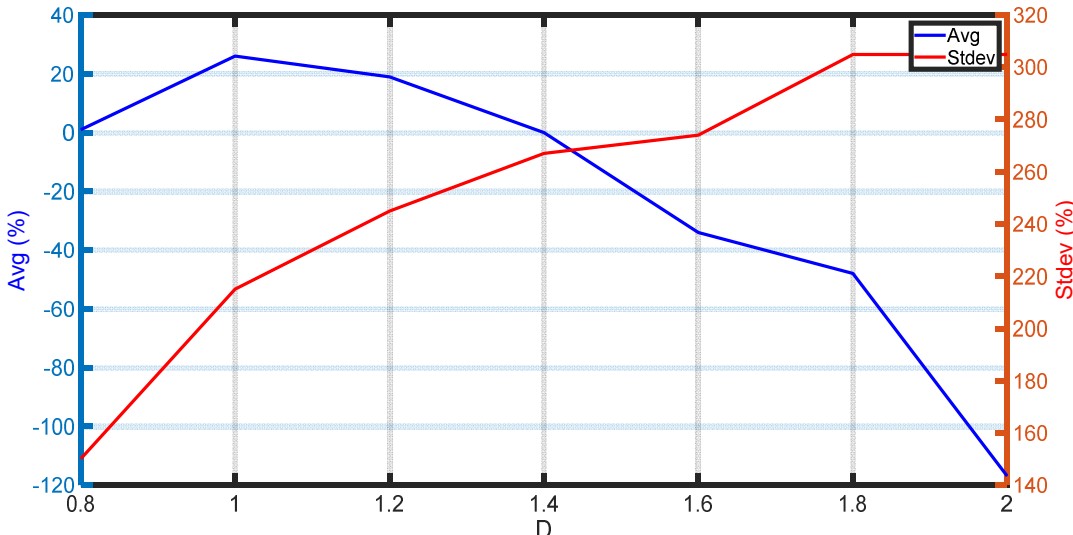

**Figure 17.** Improvement/decline in the average absolute error and standard deviation versus $D$ based on $\beta$ error resampling.

## 5. Conclusions

In this study, we present a novel ML model training strategy for phase behavior prediction, with a particular focus on improving the accuracy of calculations near criticality. We demonstrate the effectiveness of the proposed approach on the flash problem in reservoir fluids, but our findings and insights are broadly applicable to other domains involving phase separation phenomena, such as chemical engineering, materials science, and environmental engineering.

The proposed methodology is based on the principle of fine-tuning the distribution of the training data population by resampling to ensure the ML model's optimal exploitation across different input ranges. This is achieved by incorporating a physics-driven understanding of the system into the ML training process. Specifically, this technique involves fine-tuning the model's learning capacity without altering its structure, taking into consideration the impact of relevant input variables on the fluid property of interest. This technique is directly applicable to a wide range of engineering problems where an ML model utilizes an input, $\mathbf{x}$, to predict an output, $\mathbf{f}(\mathbf{x})$, with the primary focus being on the accuracy of a dependent property, $\mathbf{g}(\mathbf{f}(\mathbf{x}))$. Overall, it paves the way for the development of more precise and robust ML models for phase behavior prediction and other engineering applications where the property of interest in the design problem is a function of the ML model's originally predicted output.

**Author Contributions:** Conceptualization, E.M.K. and V.G.; methodology, E.M.K.; software, E.M.K.; formal analysis, V.G.; writing—original draft preparation, E.M.K. and A.S.; writing—review and editing, V.G.; visualization, E.M.K. and A.S.; supervision, V.G. All authors have read and agreed to the published version of the manuscript.

**Funding:** This research received no external funding.

**Data Availability Statement:** No new data were created or analyzed in this study. Data sharing is not applicable to this article.

**Conflicts of Interest:** The authors declare no conflicts of interest.

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
