# Peer review of "Enhancement of Machine-Learning-Based Flash Calculations near Criticality Using a Resampling Approach"

_computation, doi:10.3390/computation12010010_

Round 1
Reviewer 1 Report
Comments and Suggestions for Authors
In this work, the authors discussed the application of a re-sampling technique with an attempt to better predict k-value for flash calculation. The introduction is comprehensive and well written and the topic is definitely of interest to the audience of MDPI computation. However, there are a few points needed to be further discussed and addressed before my recommendation for publication.
- In the proof of concept section, random noise was used to demonstrate how the same level of prediction error at different proximity would impact the accuracy of dependent variables (line 235). The authors shall provide more details on this part (how many random noise points are generated and used for the calculation for results in Table 4). In addition, as the authors discussed, the closer the points are to critical point, the k-value is closer to 1. If the random noise of fixed amplitude instead of fixed relative amplitude is used, I would assume the noise has more impacts for the smaller k-value no matter whether it is associated with critical points. Can the authors explain more on the rationale of the experimental design?
- For the ANN model (line 268) in section 2, what are the inputs and outputs (I would assume the output is the k-value?)?
- In Figure 7, 12-16, the absolute error are plotted. However, the relative error would be of convenient for the audience to get better understanding of the model performance. If the addition of relative error would make the main context too crowded. They can be provided in the supporting information.
- According to eqn (8)-(11), the Fi would always be larger than or equal to Ni since δ would always be larger than or equal to 0. However, in Figure 9, not all the number of instance are increasing with the increase of D, can they authors provide more details on this?
- In Figure 10 and 17, the change of mean absolute error (MAE) and standard deviation are plotted. Can the authors provide the MAE, relative error and standard deviation at different D for a better demonstration of how the sampling method impact the model performance. Meanwhile, it is puzzle to me that it seems in Figure 10, the MAE is negative between 0 and 0.8 which means the sampling method deteriorates the model performance. I would strongly recommend the authors provide do a deep-dive for the value of D between 0 and 0.8 and have a in-depth discussion on why the sampling method doesn’t improve the ML model performance at the small value of D.
- In Figure 12-16, it seems the resampled training dataset would have a worse performance for classes 5-10 which corresponds to the decreased number of instance in Figure 9. Can the authors provide more explanation on why the resampled training dataset doesn’t hold up the performance for those classes compared to the base training dataset?
- Again, for table 6 and 7, it is highly recommended to provide the relative average errors for the properties discussed and covered (the authors shall consider to provide those in the supporting information).
Reviewer 2 Report
Comments and Suggestions for Authors
Thank you for providing the paper. The title " Enhancement of machine learning based flash calculations near criticality using a resampling approach," suggests a novel resampling technique of the ML models training data population is proposed which aims at the fine-tuning of the training dataset distribution and to the optimal exploitation of the models’ learn ing capacity across various flash conditions. Overall, the paper addresses a valuable topic and has the potential to contribute significantly to the field. However, some minor corrections should be made before publishing.
Suggestions for Modification:
(A) Consider providing an explanation of the software used to generate the K value. This would enhance the clarity of the methodology and improve reproducibility.
(B) The paper primarily assigns values to D based on the proximity to the critical point and loss and adjusts the data quantity for each group accordingly. It is suggested to explore the possibility of using Gaussian or other distribution methods to obtain multiple points around the critical point when generating 100,000 data points. This could add robustness to the methodology.
(C)Variable Considerations:
The main variable in the paper is pressure. To strengthen the persuasiveness of the study, it is recommended to simultaneously vary both pressure and temperature. Additionally, exploring variations in the composition ratios would contribute to a more comprehensive analysis.
Round 2
Reviewer 1 Report
Comments and Suggestions for Authors
The manuscript has been improved and most of my significant concerns have been addressed.